# The Development and Testing of a Patient Decision Aid for Individuals with Homologous Recombinant Proficient Ovarian Cancer Who Are Considering Niraparib Maintenance Therapy

Laura Hopkins [1,*], Mark Carey [2], Linda Brown [3,†], Sabryna McCrea [4], Mark Milne [4], Dawne Tokaryk [5] and Dawn Stacey [6,7]

1 Division of Oncology, University of Saskatchewan, Saskatoon, SK S7V 4H4, Canada
2 Division of Gynecologic Oncology, University of British Columbia, Vancouver, BC V6T 1Z4, Canada; mark.carey@ubc.ca
3 Ovarian Cancer Canada, Toronto, ON M5A 1E3, Canada
4 Research Facilitators, University of Saskatchewan, Saskatoon, SK S7N 5A2, Canada; sabryna13mccrea@gmail.com (S.M.); mark.milne@usask.ca (M.M.)
5 Saskatchewan Cancer Agency Patient and Family Advisory Council, Saskatoon, SK S7V 4H4, Canada; dhtokaryk@sasktel.net
6 School of Nursing Science, University of Ottawa, Ottawa, ON K1H 8M5, Canada; dawn.stacey@uottawa.ca
7 Centre for Implementation Research, Ottawa Hospital Research Institute, Ottawa, ON K1H 8L6, Canada
* Correspondence: laura.hopkins@saskcancer.ca
† This author deceased after decision aid was finalized and before manuscript written.

**Abstract:** New treatments for ovarian cancer are available that require trade-offs between progression-free survival and quality of life. The aim of this study was to develop a decision aid for patients with homologous recombinant proficient (HRP) tumors, as the benefit–harm ratio of niraparib needs consideration. This decision aid was created with a systematic and iterative development process based on the Ottawa Decision Support Framework. The decision aid was user-tested for acceptability, usability, and comprehensibility using a survey completed by a sample of patients with ovarian cancer and oncologists. This decision aid follows the International Patient Decision Aids Standards (IPDAS) criteria in its development. User-test respondents (n = 13 patients; 13 physicians) reported that the decision aid used language that was easy to follow (69% patients; 85% physicians), was an appropriate length (69% patients; 62% physicians) and provided the right amount of information (54% patients; 54% physicians). Most respondents (92% patients; 62% physicians) would recommend this decision aid for HRP patients considering niraparib. This is the first decision aid for patients with HRP ovarian cancers who are considering niraparib maintenance therapy. It is available on-line and is being further evaluated in a pragmatic clinical trial in Saskatchewan.

**Keywords:** decision aid; ovarian cancer; niraparib

## 1. Introduction

Homologous recombinant deficiency (HRD) is a tumor characteristic defined by the inability to repair double-stranded DNA breaks. In ovarian cancers, we identify HRD by looking at specific tumor biomarkers: either individual mutations (for example BRCA1 or BRCA2) or measurable and structural rearrangements in DNA. As such, we may identify ovarian cancer patients as being homologous recombinant deficient or proficient. Patients whose tumors are HRD respond very well to chemotherapy, whereas for patients whose tumors are homologous recombinant proficient (HRP), the response is poor. HRD testing also provides diagnostic information for patients with ovarian cancer that oncologists can use to personalize treatment options. For example, patients with HRD tumors are more likely to benefit from a new class of drugs called poly(ADP-ribose) polymerase inhibitors (PARPis) than patients with HRP tumors. Thus, tumor testing can identify

patients who can get the maximum benefit from PARPis, following initial platinum-based chemotherapy treatments.

The results of a randomized, placebo-controlled trial (PRIMA) have shown that for patients whose tumors were HRD, a PARPi called niraparib was associated with a median progression-free survival time of 9–20 months compared to a placebo [1]. For patients whose tumors were HRP, the median prolongation in progression-free survival was 3 months compared to a placebo [2]. Despite this differential benefit, all patients that are 'in response' to their front-line chemotherapy treatments qualify for maintenance PARPi without consideration of HRD tumor status. Tumor testing is a precision oncology approach that allows clinicians to be very specific about the amount of benefit that a patient can expect from PARPi treatment. There is no standard of care or approval for tumor testing in Canada, although in the Unites States and many other countries, HRD testing is routinely carried out.

When any new drug is started by a patient, there are generally several weeks during which side effects are seen and experienced. LaFargue et al. published an overview of PARPi toxicity and provided evidence that this period lasts up to 3 months in the case of PARPis [3]. As such, there is a need to weigh the potential benefits versus harms of PARPi maintenance, especially for HRP patients, since the benefits versus harms are about even. We have opened a precision oncology pragmatic trial for all advanced-stage ovarian cancer patients that will provide real-world evidence for the value of tumor testing. Given the trade-off between benefit and risk for patients identified as being HRP, we felt a decision aid would help present evidence-based information regarding options so that patients could decide whether PARPi treatment was right for them in the context of their personal priorities and values.

Decision aids are clinical resources that can be used by patients to help reduce their decisional conflict and enhance communication between patients and physicians. They are especially useful when there is clinical uncertainty or equipoise in terms of recommending a given treatment. Decision aids are associated with higher rates of patient satisfaction with their final decision and have been shown to improve patient understanding of therapy-associated risks, benefits, and how their decision aligns with their personal values [4]. According to the criteria outlined by the International Patient Decision Aid Standards (IPDAS) collaboration, a decision aid must meet minimum requirements including clearly stating the decision that must be made and presenting all potential options in the context of associated risks and benefits [5]. To our knowledge, there is no decision aid focused on the PARPi called niraparib for women with ovarian cancer whose tumors are tested and found to be HRP. The aim of this study was to create, field test, and implement the use of a decision aid resource for these patients. Validation/beta testing of the decision aid with new patients who are HRP and facing the option of niraparib maintenance therapy will be performed in conjunction with our precision oncology trial.

## 2. Materials and Methods

### 2.1. Study Design

Our patient decision aid was developed using the Ottawa Patient Decision Aid Development e-Training template. This e-training is based on the Ottawa Decision Support Framework [6,7], and the structure and development process was based on the International Patient Decision Aids Standards (IPDAS) [8]; see Figure 1. The hypothesis underlying this framework is that decision support interventions (such as our decision aid) will address patients' decisional needs and result in patient-driven treatment decisions with reduced decision conflict. The Ottawa template has been evaluated in 24 randomized controlled trials including a variety of clinical decisions and has been shown to improve the quality of decisions and results in reduced decisional conflict for patients [9]. The IPDAS collaboration uses an evidence-based framework to improve the content, development, implementation, and evaluation of patient decision aids. The process of developing a patient decision aid follows 7 basic steps: (1) determining the target population and treatment options,

(2) establishing a steering committee, (3) performing a literature review to understand the appropriate options and outcomes of the decision aid, (4) creating a prototype using the Ottawa template, (5) alpha testing (field testing) the prototype among patients and physicians to assess readability and accessibility, (6) using the results of alpha testing to make adjustments, and finally, (7) beta testing the decision aid to quantify decision-making outcomes. We are now completing the beta testing in our precision pragmatic trial.

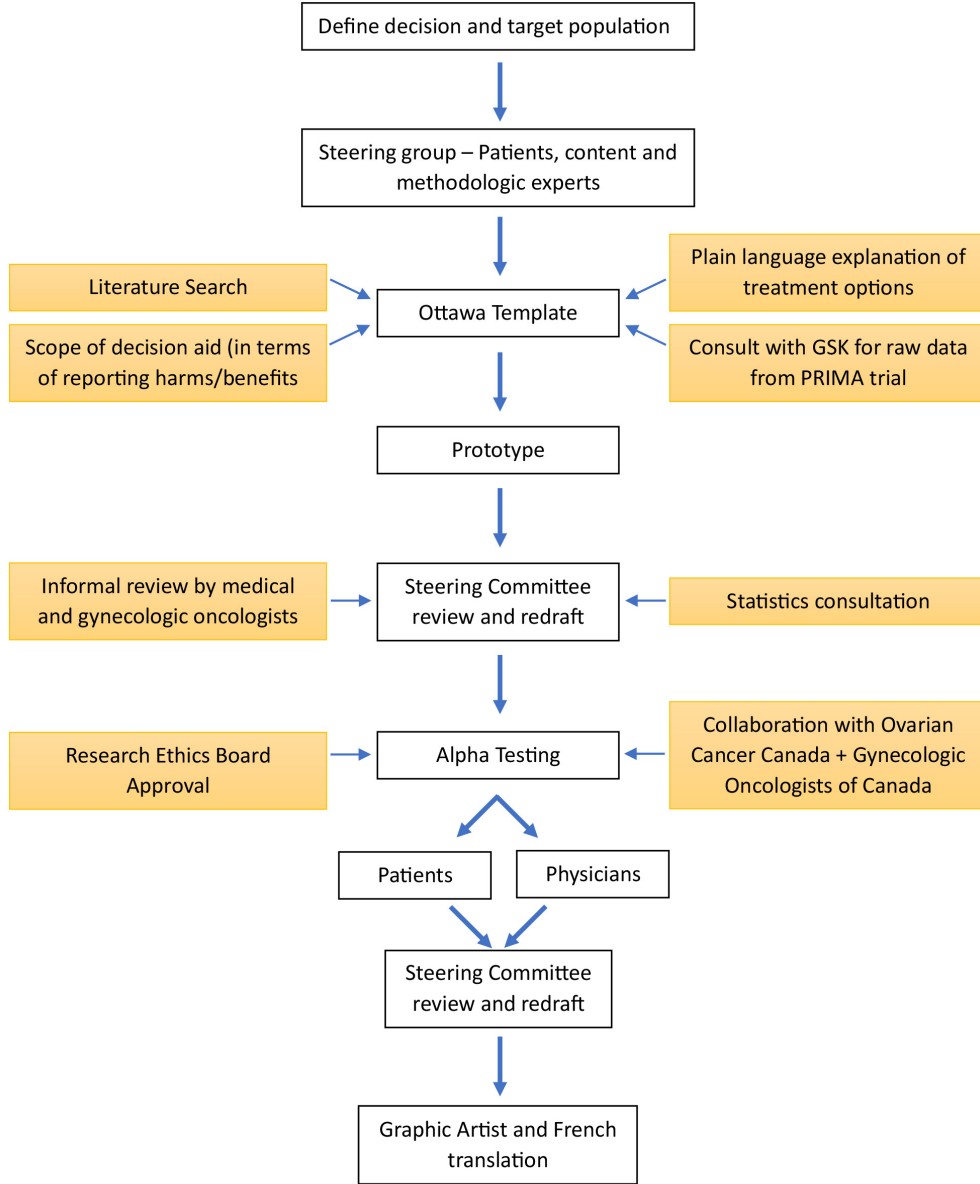

**Figure 1.** Structure and development process for the decision aid.

### 2.2. Population and Treatment Options

The target group for our decision aid is patients with advanced ovarian cancer who have completed initial treatment and meet the eligibility criteria for PARPi treatment. The results of tumor testing must reflect that the patient is HRP in order to be eligible to use the decision aid. Patients who are homologous recombinant deficient or who do not meet eligibility for PARPi maintenance therapy according to existing standards are not the target population. Standard treatment options for the HRP population include taking niraparib or not taking niraparib. The steering committee discussed other possible therapeutic options (i.e., clinical trials) and concluded that none were consistently available to this population and thus could not be listed within the decision aid as an additional option for care.

*2.3. Steering Committee*

A committee of experts was assembled including gynecologic oncologists, an international expert in decision aid development, research associates, and ovarian cancer patients. These patient partners provided unique insights and experiential leadership and guidance in the content and presentation of information (including structure, layout, terminology) using their lived experience with cancer.

*2.4. Literature Review*

Benefits and harms data were collected through a review of the PRIMA trial, the niraparib product monograph [10], and the Common Terminology Criteria for Adverse Events [11]. The PRIMA trial was the primary resource for benefits and harms data as it is the only clinical trial available that examines niraparib use in the front-line maintenance setting. The 3.5-year follow-up PRIMA data were used to quantify the benefit for the HRP group.

*2.5. Prototype Development*

A prototype of the decision aid was developed using the Ottawa template and then was reviewed across ten meetings of the full steering committee and four additional iterative revisions following initial alpha testing. The decision aid was written in plain language with an aim of achieving a SMOG (Simple Measure of Gobbledygook) reading level of 7 or lower so that most patients with different levels of education, literacy, and ability could use the document [12]. The decision aid includes intuitive figures and colors to represent the incidence and level of severity of all described outcomes in individuals to facilitate patient understanding. We contacted Glaxo Smith Klein and requested the PRIMA raw data so that a range of benefit could be calculated and presented in the decision aid with the idea that it would be easier for patients to understand. GSK responded that only data available in the public domain were accessible. The harms that are listed in the decision aid include all those that occurred with more than 10% frequency in PRIMA: thrombocytopenia, anemia, nausea, fatigue, neutropenia, constipation, headaches, insomnia, dyspnea, vomiting, decreased appetite, hypertension, acute kidney injury, and leukemia. A visual diagram is provided in Figure 2 to show how the numbers of patients affected by each outcome are represented in the decision aid. The outcomes are represented by groups of 100 figures which are shaded according to severity (grades 3 to 4 are in orange shading; grades 1 to 2 are in teal shading) to indicate the number of patients expected to have each outcome. We included knowledge testing questions such as "Which option has the highest chance of giving me more time before my next treatment?" in the decision aid to help confirm the patient's comprehension. These questions are followed by an answer key at the bottom of the page so that patients may verify their answers and assess their understanding. The decision aid includes the SURE test, a validated 4-item instrument used to screen for Decisional Conflict [13]. A score of 4/4 indicates no decisional conflict, whereas lower scores can help the oncologist determine whether additional discussion is needed.

## Benefits

In the future, you will be offered more chemotherapy. Taking Niraparib may delay this offer of chemotherapy by about 3 months. There is no data that Niraparib can prolong your life.

## Harms/Side Effects

Like all drugs, Niraparib has a number of side effects. Some of these are serious. Understanding the side effects of this drug will help you decide whether Niraparib is right for you.

### Icon information

- Teal means mild to moderate [3]
- Orange means severe to life-threatening [3]

**Low Platelet Numbers**

- If 100 people take Niraparib, 66 will have low platelet numbers (Thrombocytopenia).
- If 100 people do not take Niraparib, 5 will have low platelet numbers (Thrombocytopenia).
- Low platelet counts range from just a low number on your bloodwork to a risk of bleeding that might require a platelet transfer.
- The average time until symptoms of low-platelet numbers start is 22-23 days. Symptoms last from several days to several weeks. Symptoms improve after the third month of treatment.

**Niraparib**

66 have low platelet numbers, 34 do not

**No Niraparib**

5 have low platelet numbers, 95 do not

**Figure 2.** Visual diagram depiction of benefits/harms of niraparib contained in the decision aid.

### 2.6. Alpha Testing

The alpha testing methodology was approved by the University of Saskatchewan Behavioural Research Ethics Board, certificate number 3772, activated 10 January 2023. Alpha testing involved surveying a sample of both patients and physicians to assess the readability, usability, and accessibility of this decision aid. This involved scrutinizing various aspects of the decision aid including length, accessibility of content, balanced and unbiased information, and its effectiveness in addressing the target decision. The survey was a nine-question, mixed-methods survey which was distributed via SurveyMonkey. The patients and physicians approached to complete this survey were asked to review the decision aid prototype before completing the survey. Eligible participants were patients with ovarian cancer and gynecologic and medical oncologists who treat ovarian cancer patients. Ovarian Cancer Canada sent the survey link and the decision aid to all twenty members of their Patient Partners in Research team. The Gynecologic Oncologists of Canada posted the survey link and the decision aid in their members e-newsletter (sent out to 207 retired or active gynecologic and medical oncologists who treat ovarian cancer) which was distributed weekly during the study period. The survey was open to volunteer response for 4 weeks and then closed. The alpha testing step is critical in decision aid development as it provides an opportunity to receive feedback from multiple stakeholders and perspectives intended to improve the final product. Suggestions and comments from alpha testing were reviewed by the steering committee and the decision aid was updated. In terms of a sample size for reviewer feedback, this has been established by Guest et al., who attested in 2006 that 'saturation. . . has become the gold standard by which diversity samples

are determined in health science research' [14]. Upon review of participant responses, we made a team decision that data saturation was reached with a sample size of eight for each of the patient and physician groups. This means that no new comments were being provided by participants that developed our decision aid concept any further after this number. Where there was lack of consensus in terms of whether certain suggestions for change would be incorporated, the patient members of the steering committee had the final say. The final draft was then forwarded to a graphic artist team to enhance the visuals and translated into French.

## 3. Results

We received a total of 26 responses from the alpha testing process. Thirteen responses were from patients and thirteen responses were from physicians. Numerical feedback showed that the two groups felt that the language used in the decision aid was easy to follow (69% patients; 85% physicians), was an appropriate length (69% patients; 62% physicians), and provided the right amount of information (54% patients; 54% physicians). For respondents who disagreed that the decision aid provided the right amount of information, patients thought more detail about HRD testing was needed and physicians felt that the harms section was too long and overly detailed. Most respondents (92% patients; 62% physicians) felt they would recommend this decision aid for HRP patients considering niraparib. There were differences in opinion as to whether the decision aid was well balanced, with some physicians and patients feeling there was too much detail and others feeling there was too little detail; as such, only 69% of patients and 54% of physicians thought the decision aid was balanced. The steering committee struggled with this feedback in terms of the conflicting responses; the patient partners made the final decision, which was to keep the length of the decision aid as it was and to try to incorporate requested details without changing the overall reading level. When the details increased, the reading level increased, and so, it was not possible to please everyone. Respondents were in general agreement (85% patients; 85% physicians) that the benefits and harms section was easy to follow.

Narrative feedback from both groups of respondents was overall positive and provided additional suggestions which were taken into consideration and implemented in the decision aid where possible. Physician suggestions were centered toward the accessibility for patients of different education levels and cognitive abilities, specifically in terms of the ability of patients to interpret complex concepts such as progression-free survival and the meaning of 'placebo'. Patient suggestions were centered around content, with many respondents suggesting the removal of the SURE test and further explanation of progression-free survival, PARPis, and HRD testing. There was mixed response to the visual diagrams representing risks of niraparib, with some respondents' commenting that the order should be different. Our patient steering committee members made the final decision regarding the order (most common to least common with serious/life threatening presented ahead of mild/moderate followed by none) for the various included side effects of niraparib.

The results from both groups of alpha test respondents were used to modify the decision aid prototype to create the final document. Major changes that were made included explaining progression free survival, PARPi, and the HRD testing/genetic changes in greater detail. In response to feedback regarding the presentation of harms (the use of shaded/non-shaded faces), we hired a graphic artist to improve the visual diagrams. The graphic artists also suggested changes in color scheme and font to make the decision aid easier to read.

The Patient Decision Aid on Niraparib Treatment—Patient Survey (Supplementary File S1) is available free of charge (in English and French) and is accessible through the International Patient Decision Aid A to Z Inventory (https://decisionaid.ohri.ca/AZsumm.php?ID=2059, accessed on 15 February 2024) [15].

## 4. Discussion

Our team developed a decision aid to support women with advanced ovarian cancer considering niraparib maintenance therapy. It is currently being used by HRP patients in our precision oncology pragmatic trial. The alpha test results revealed generally positive feedback about the decision aid, and the participants' suggestions were used to strengthen the presentation of additional information on niraparib in the decision aid. We received feedback from thirteen physicians and thirteen patients, reaching data saturation at eight responses for both groups. In qualitative research, the appropriate sample size is a function of the purpose of the study, the complexity, range, and distribution of experiences or views of interest, rather than standard statistical parameters used in quantitative research (i.e., *p*-values, power calculations). Indeed, Francis et al. analyzed qualitative data from several studies and concluded with confidence that setting a minimum sample size of thirteen is 'very likely' to capture almost all the beliefs relating to attitude and subjective norms [16]. Guest et al. performed a similar analysis and concluded that twelve is a sufficient sample size for interview studies assessed for emergent themes [14]. Results from our alpha testing were similar to other decision aids developed using the Ottawa template [17–20]. Ivankovic et al. developed and alpha tested their decision aid on extended thromboprophylaxis following major pelvic surgery in 2022 [20]. Physician feedback was split (50/50) in terms of the length of the decision aid and the amount of information contained.

Our decision aid was designed to address an unmet need by HRP patients facing the option of niraparib maintenance therapy. Niraparib maintenance therapy for HRP patients offers benefits which need to be traded-off against the risk of important side effects at a time when patients would otherwise feel entirely well. From the literature, most side effects occur within the first few weeks of therapy and then resolve by twelve weeks, meaning that the amount of progression-free survival benefit is about equal to the amount of time managing what can be problematic and serious side effects. Any therapy that potentially decreases quality of life without offering an overall survival benefit needs careful consideration. Niraparib only offers a delay to starting second-line chemotherapy. Although no difference in quality of life was detected from the PRIMA trial (comparing a placebo and niraparib), the study methodology did not incorporate any quality of life instruments designed with PARPi toxicity measurement in mind, nor was the frequency of assessment of quality of life sufficient to capture a decrease in quality of life when it could have occurred. As such, there is uncertainty for many oncologists about the decision to recommend niraparib in the HRP setting.

Giving patients unbiased and high-quality information so they can understand the benefits versus harms of a given therapy and encouraging them to make a decision that is right for them is the very best we can do as physicians. Given the challenge of decisions where benefits are about equal to harms, it is vital that patients have relevant information presented in a manner that is known to be effective for their decision support. Decision aids address this goal and allow patients time to process and consider the implications of their decision before discussion with their physician. An important element of every decision aid is the explicit values clarification exercise that helps patients better understand what is important to them so that they make the best possible decision. Barriers to patient participation in shared decision making have been previously studied and found to include lack of knowledge of options and lack of knowledge of their values [21]. Decision aids are an effective intervention for overcoming these barriers [22]. It is our hope that this decision aid will promote shared decision making and help foster patient-centered care. While decision aids allow patients to become active agents in their care, they do not replace consultation with a physician. Rather, they may assist with patient preparation for the discussion of their options for care and then may be used concurrently in the consultation. In this way, a decision aid will assist with the review of a patient's understanding of their options and their values associated with those options and even ensure that critical information needed for that decision has not gone uncommunicated (i.e., niraparib in the maintenance setting for HRP patients has no survival benefit). There are also considerations

that may affect a decision which must be discussed with a physician such as co-morbidities that can impact the safety of treatment and/or the availability and eligibility for other options such as a clinical trials protocol for HRP patients.

There are several strengths to consider in our decision aid development process. Firstly, we had the meaningful involvement and membership of two patient partners in our steering committee. These patient partners had a governing voice throughout the development and iterative final prototype review process. Previous research has shown that when patients are involved in research, there are improved outcomes [23]. Our alpha testing exercise included anonymous ovarian cancer patients from across Canada with no connection to the research team. Similarly, the clinicians who evaluated the decision aid were also anonymous, representing sites across Canada. Coulter et al. published a review paper on the development process for decision aids [24]. They found that only about half of the published reports of decision aid development methods were alpha tested by patients, and even fewer were alpha tested by clinicians not involved in the development process. They also found that the distribution strategy for the alpha testing was rarely described by authors. A potential limitation of this paper is that we cannot yet report on the results of beta testing. Beta testing is the final step which validates a decision aid, although the importance of this step is controversial. Previous articles that outline patient decision aid development suggest beta testing is unnecessary when a validated process (such as ours) is used for development [24,25]. Experience with practical use of our decision aid within our precision oncology pragmatic trial will provide utility and value data.

## 5. Conclusions

This decision aid was developed for patients with HRP advanced ovarian cancers who have undergone tumor testing, qualify for PARPi treatment, and are now considering taking niraparib. This decision aid is a structured patient information resource that has been shown to be acceptable to patient partners and oncologists. This decision aid addresses decisional needs for ovarian cancer patients that are not met by current resources and will help to ensure that patients are well informed when making this decision with their oncologist.

**Supplementary Materials:** The following supporting information can be downloaded at: https://www.mdpi.com/article/10.3390/curroncol31030107/s1, File S1: Patient Decision Aid on Niraparib Treatment—Patient Survey.

**Author Contributions:** Conceptualization L.H.; methodology D.S.; formal analysis L.H., S.M.; resources L.H., L.B., D.T.; data curation L.H., S.M.; writing -original draft preparation, LH, S.M.; writing—review and editing M.C., D.S., M.M., L.B., D.T; supervision L.H.; project administration L.H., M.M.; funding acquisition L.H. All authors (L.B. deceased prior to manuscript draft being written) have read and agreed to the published version of the manuscript.

**Funding:** Funding was provided by the Province of Saskatchewan as represented by the Minister of Health to OVARIAN CANCER CANADA in support of the OCAN research initiative. Funding contribution was also provided by the SASKATCHEWAN HEALTH RESEARCH FOUNDATION (research connection grant 6282).

**Institutional Review Board Statement:** The study was conducted in accordance with the Declaration of Helsinki and approved by the Institutional Research Ethics Board of the UNIVERSITY OF SASKATCHEWAN (protocol number 3772; date of approval 10 January 2023).

**Informed Consent Statement:** Informed consent was obtained from all subjects involved in this study.

**Data Availability Statement:** Data supporting reported results can be found by contacting the corresponding author.

**Acknowledgments:** We would like to acknowledge the support of the Patient and Family Advisory Council of the Saskatchewan Cancer Agency, Ovarian Cancer Canada Patient Partners in Research, and the Gynecologic Oncologists of Canada.

**Conflicts of Interest:** Hopkins is on the Ovarian Cancer Canada Governing Council. None of the remaining authors have any conflicts of interest to declare. Ovarian Cancer Canada circulated the alpha test survey to members of its 'Patient Partners in Research' group for consideration of anonymous participation.

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
