# Peer review of "The Development and Testing of a Patient Decision Aid for Individuals with Homologous Recombinant Proficient Ovarian Cancer Who Are Considering Niraparib Maintenance Therapy"

_curroncol, doi:10.3390/curroncol31030107_

Round 1

Reviewer 1 Report

Comments and Suggestions for Authors

The aim of the work "Development and Testing of a Patient Decision Aid for Individuals with Homologous Recombinant Proficient Ovarian Cancer Who Are Considering Niraparib Maintenance Therapy" submitted to Curreny Oncology was to create, test in practice and implement a resource to support decision-making regarding the use of an adjunctive PARPi treatment called niraparib in ovarian cancer patients whose tumors were tested and found to be HRP. It is important for deciding whether to use PARPi therapy for ovarian cancer patients with homologous recombinant proficient (HRP) tumors. There is no standard of care or approval for tumor testing in Canada, although in the Unites States and many other countries, HRD testing is routinely done.

The question is clearly defined and the manuscript is written quite clearly. The topic is important from the point of view of patients with ovarian cancer who have Homologous Recombinant Proficient (HRP) tumors. The Figures are clear and legible.

Minor points:

1. Results: I'm missing how many people completed the surveys (Figure 2 shows 200? people and I don't have such numbers in the text).

2. Conclusions: I'm missing detailed information about the patient survey. How many took part in it. Were only side effects taken into account?

Author Response

  1. Regarding Figure 2. This Figure is a depiction taken directly from the decision aid to show readers how we displayed the side effects information for patients.  The benefit of niraparib in the decision aid is expressed as a simple statement and isn’t much to look at in terms of an included figure.   Figure 2 shows what would happen to 100 people if they took niraparib or if they did not.  So there are not 200 people in this study.  There is language in Figure 2 already that I believe makes this distinction clear.
  2. The patient survey will be attached.  The manuscript document already states that we had 13 physician respondents and 13 patient respondents.  For those who want to view the decision aid, it has been included already in the text with a ‘hot link’ which goes directly to the page on the OHRI website.  The decision aid is freely available there in both English and French for anyone who wants to examine it in more detail or use it.

Reviewer 2 Report

Comments and Suggestions for Authors

The manuscript submitted for review concerns an interesting and important aspect of the treatment of patients with ovarian cancer. It is co-making decisions in therapeutic options based on tools supporting such decisions. The manuscript is clearly written, pleasant to read, and arouses interest.

My little comments.

1. On what basis was the size of the research groups determined?

2. Wouldn't it be worth comparing some of the obtained data with the control group?

3. Information about PFS in patients using niraparib does not clearly state that this drug prolongs PFS. (l.50-52)

4. Figure 1 contains abbreviations without explanation.

The manuscript is suitable for publication after minor additions and corrections.

Author Response

  1. The sample size for the survey for the decision aid feedback is elaborated in the yellow highlighted text. I also included two new references to support the fact that 12 or 13 responses are considered sufficient.  I also included some language from our experience which suggested to us that a sample size of 8 was sufficient to reach ‘data saturation’.  I added an explanation of what data saturation means.  I think it makes the paper much stronger and thank-you for this question.
  2. There is no control group in this survey. We survey stakeholders only; the physicians and the patients.  The data that compares their responses (similarities and differences) is already included in the ‘results’ section.  I re-examined all the responses yesterday and there isn’t anything else I can really ‘do’ with these data.
  3. I included specific language to address ‘prolongation’ of PFS. Thank-you.
  4. Figure 1 – REB, changed to Research Ethics Board. Thank-you.

Reviewer 3 Report

Comments and Suggestions for Authors

Laura Hopkins et al. developed and compiled notes on a patient decision aid designed for individuals with HRP advanced ovarian cancers who have undergone tumor testing, are eligible for a PARPi, and are contemplating the use of niraparib as maintenance therapy. The decision aid serves as a structured patient information resource and found to be acceptable to patients and gynecologic oncologists.

The manuscript is comprehensive, covering various aspects of patient decision aids such as study design, literature review, steering committee, alpha testing, and the harms associated with niraparib treatments. However, there are some minor unresolved issues.

Comments:

It is noted that in Canada, niraparib is the only PARPi currently used for ovarian cancer treatment. Including a table detailing patient age, level of education, type of chemotherapy/surgery received, and prolonged tumor growth free survival in those who receive niraparib as maintenance therapy would strengthen the article.

Additionally, the authors could consider discussing the average ovarian cancer patient survival rate after receiving niraparib or no niraparib treatment to provide a more comprehensive analysis which may benefit readers.

Author Response

Yes, niraparib is the only PARPi approved in Canada for the proficient setting.  There is no possibility for me to include a table the way you suggest as the data is not available ‘in the public domain’ via the PRIMA trial.  I read PRIMA over again today.  At no point did they collect education level. The ‘type of chemotherapy’ was 6 -9 cycles of Carbo/Taxol with no differentiation according to HRP patients who took niraparib versus placebo.  Similarly, age is not available for the HRP group vs placebo and the progression free survival data that we have in this manuscript is the most up-to-date information available today on niraparib maintenance.  It is taken from a referenced paper that was published subsequent to PRIMA with 3.5 year follow-up.  This is described already in the manuscript text. The PRIMA group of authors do not present any number beyond the 3 months delay in progression/relapse when patients take niraparib vs those who took placebo for the HRP group.   The only intended users of the decision aid are the HRP group, so presentation of other data results from PRIMA (beyond the basics) does not add value nor strength to this manuscript in my view.

There is no survival benefit for patients who are HRP who take niraparib from the available data/PRIMA trial nor longer-term follow-up reports.  Delaying relapse/progression is only a surrogate marker for overall survival and it is imperfect.  This is a difficult concept. Basically, a primary outcome of PFS can be established with smaller sample sizes of patients.  To show survival benefit of any therapy requires huge numbers of patients and is cost-prohibitive.  So, here we are.  Women with HRP do not respond to any chemotherapy drug as a generalization; starting chemo in the recurrent setting is only going to (hopefully) improve quality of life by keeping the disease controlled.  Disease control is a short-lived phenomenon in the HRP setting.  The disease comes back, is resistant to treatment and then the patient unfortunately dies.  Niraparib doesn’t alter that outcome in any way.  Delaying onset of second-line chemo is not synonymous with response to second-line chemo.  This is why survival is  not impacted.